# Integrating Regenerative Medicine in Chronic Wound Management: A Single-Center Experience

**DOI:** 10.3390/biomedicines13081827

**Published:** 2025-07-25

**Authors:** Stefania-Mihaela Riza, Andrei-Ludovic Porosnicu, Patricia-Alina Cepi, Sorin Viorel Parasca, Ruxandra-Diana Sinescu

**Affiliations:** 1Department 11, Discipline Plastic and Reconstructive Surgery, Bucharest Clinical Emergency Hospital, University of Medicine and Pharmacy Carol Davila, 050474 Bucharest, Romania; stefania-mihaela.riza@drd.umfcd.ro (S.-M.R.); sorin.parasca@umfcd.ro (S.V.P.); ruxandra.sinescu@umfcd.ro (R.-D.S.); 2Department of Plastic Surgery and Reconstructive Microsurgery, Elias Emergency University Hospital, 011461 Bucharest, Romania; cepipatricia@gmail.com; 3Emergency Clinical Hospital for Plastic, Reconstructive Surgery and Burns, 218 Grivitei Street, 010713 Bucharest, Romania

**Keywords:** chronic wounds, platelet-rich plasma (PRP), autologous fat grafting, regenerative medicine, adipose-derived stem cells (ADSCs), skin grafting, wound healing

## Abstract

**Background**: Chronic wounds represent a persistent clinical challenge and impose a considerable burden on healthcare systems. These lesions often require multidisciplinary management due to underlying factors such as microbial colonization, impaired immunity, and vascular insufficiencies. Regenerative therapies, particularly autologous approaches, have emerged as promising strategies to enhance wound healing. Adipose tissue-derived stem cells (ADSCs) and platelet-rich plasma (PRP) may improve outcomes through paracrine effects and growth factor release. **Methods**: A prospective observational study was conducted on 31 patients with chronic wounds that were unresponsive to conservative treatment for over six weeks. Clinical and photographic evaluations were employed to monitor healing. All patients underwent surgical debridement, with adjunctive interventions—negative pressure wound therapy, grafting, or flaps—applied as needed. PRP infiltration and/or autologous adipose tissue transfer were administered based on wound characteristics. Wound area reduction was the primary outcome measure. **Results**: The cohort included 17 males and 14 females (mean age: 59 years). Etiologies included venous insufficiency (39%), diabetes mellitus (25%), arterial insufficiency (16%), and trauma (16%). Most lesions (84%) were located on the lower limbs. All patients received PRP therapy; five underwent combined PRP and fat grafting. Over the study period, 64% of the patients exhibited >80% wound area reduction, with complete healing in 48.3% and a mean healing time of 49 days. **Conclusions:** PRP therapy proved to be a safe, effective, and adaptable treatment, promoting substantial healing in chronic wounds. Autologous adipose tissue transfer did not confer additional benefit. PRP may warrant inclusion in national treatment protocols.

## 1. Introduction

Chronic wounds are lesions that fail to progress through the normal phases of healing, namely hemostasis, inflammation, proliferation, and remodeling, within an expected timeframe [1,2]. They are commonly defined as wounds that do not exhibit a ≥50% reduction in size after four weeks or that fail to achieve complete epithelialization within 90 days [3]. The estimated prevalence of chronic wounds in Europe and the United States is approximately 2.5%, a figure that continues to rise, placing an increasing economic and clinical burden on healthcare systems while significantly impairing patients’ quality of life and social functioning [3,4,5]. Although mostly prevalent in older adults (>65 years), recent data indicate an increasing incidence in younger populations, primarily due to the rising prevalence of comorbidities such as diabetes mellitus, obesity, metabolic syndrome, and peripheral vascular disease [1,4]. These conditions serve as key risk factors that perpetuate delayed wound healing. The lower extremities represent the most frequently affected anatomical site. The principal etiologies include diabetic foot ulcers, pressure injuries, chronic venous insufficiency, and, to a lesser extent, peripheral arterial disease. Less common causes involve traumatic wounds, malignancies, storage diseases (e.g., amyloidosis), immunosuppressive therapy, and systemic connective tissue disorders (e.g., systemic sclerosis) [6].

The chronicity of these wounds is driven by both intrinsic and extrinsic factors. Locally, ischemia, necrosis, infection, and foreign bodies hinder healing. Systemically, poorly controlled diabetes, renal insufficiency, immunosuppression, malnutrition, and radiation exposure contribute to impaired tissue regeneration [7,8]. One of the critical pathophysiological mechanisms is sustained bacterial colonization. While all open wounds are colonized to some extent, chronic wounds provide an optimal environment for bacterial proliferation, particularly under conditions of tissue hypoxia. Oxygen-dependent enzymes such as myeloperoxidase, which is important for leukocytes’ antimicrobial function, are impaired in low-oxygen microenvironments, reducing immune efficiency and promoting infection [9,10]. The accumulation of necrotic tissue and protein-rich exudate further sustains microbial growth and facilitates biofilm formation, which acts as a physical and chemical barrier to both the host immune response and antimicrobial therapies [11,12]. This results in a perpetuating inflammatory loop, characterized by increased cytokine and protease activity, extracellular matrix degradation, and impaired keratinocyte migration, all of which halt progression toward wound closure.

The management of chronic wounds is complex, often requiring multidisciplinary approaches and the integration of specialized care [13,14,15]. Treatment algorithms vary across healthcare systems depending on resource availability, infrastructure, and practitioner experience [16]. Patient-specific factors including comorbidities, education level, socioeconomic status, treatment adherence, and lifestyle habits such as smoking further modulate outcomes [17,18].

Emerging regenerative therapies, including platelet-rich plasma (PRP) and adipose-derived stem cell (ADSC) applications, have gained attention for their potential to enhance tissue regeneration and modulate inflammation. These autologous interventions may serve as valuable adjuncts in the management of chronic wounds, particularly when conventional treatments fail. Platelets are a rich source of alpha granules, which release a complex array of growth factors, cytokines, chemokines, and membrane proteins upon activation [19]. These mediators play essential roles in various physiological processes including hemostasis, immune regulation, and tissue repair, and are important to the transition of wounds from inflammation to regeneration. Key components include Vascular Endothelial Growth Factor (VEGF) and Hepatocyte Growth Factor (HGF), which promote angiogenesis; Epidermal Growth Factor (EGF) and Insulin-like Growth Factor (IGF), which stimulate epithelial proliferation and cellular metabolism; Platelet-Derived Growth Factor (PDGF) and Fibroblast Growth Factor (FGF), which enhance fibroblast activity and vascularization; and Transforming Growth Factor Beta (TGF-β), which modulates extracellular matrix production and inflammation [20,21].

The regenerative mechanisms attributed to autologous fat transfer are multifactorial. It enhances angiogenesis and microvascular remodeling, modulates the immune response by recruiting anti-inflammatory M2 macrophages, and secretes antioxidants that mitigate oxidative stress. Additionally, it supports extracellular matrix remodeling by regulating fibroblast activity, exerts anti-apoptotic and tissue-protective effects, and promotes lymphangiogenesis, thereby reducing edema and improving local immune regulation [22].

In the present study, we assessed the clinical effectiveness of PRP and autologous fat grafting, individually and in combination, in patients with chronic wounds of varied etiologies. All participants underwent serial debridement and standardized local wound care, followed by individualized treatment involving PRP, adipose tissue transfer, or both. The primary outcome was the percentage reduction in wound surface area, used to evaluate the therapeutic response over time.

## 2. Materials and Methods

### 2.1. Study Design and Population

This prospective observational study was carried out at a specialized wound care unit between September 2024 and March 2025. The aim was to evaluate the clinical effectiveness of autologous fat grafting and platelet-rich plasma (PRP) therapy in promoting healing in chronic wounds of various etiologies. A total of 35 patients with chronic wounds persisting for more than six weeks were included. All patients underwent serial surgical debridement, received standardized local wound care, and were subsequently treated with PRP and/or autologous fat transfer.

Patients were divided into two primary groups depending on whether they underwent surgical coverage with a split-thickness skin graft (STSG) or received conservative treatment without grafting, depending on the wound characteristics and clinical indications.

Graft-assisted group: Patients treated with PRP/fat grafting plus a skin graft;Regenerative-only group: Patients treated with PRP/fat without grafting.

Inclusion criteria comprised the following:(1)Chronic wound duration of >6 weeks;(2)Absence of systemic infection;(3)ASuitability for surgical debridement.

Exclusion criteria included the following:(1)Signs of acute infection with systemic involvement;(2)Coagulation disorders;(3)Contraindications for autologous blood or fat harvesting.

### 2.2. Data Collection and Clinical Evaluation

Clinical and demographic data were collected, including age, sex, BMI, smoking status, presence of comorbidities (e.g., diabetes, chronic venous insufficiency, peripheral arterial disease), wound etiology, location, surface area, and age of lesion. Wound progression was monitored through serial measurements and photographic documentation. The primary endpoint was the percentage reduction in wound surface area, with secondary endpoints including complete wound healing, infection rate, antibiotic use, and the need for adjunctive therapies such as negative pressure wound therapy (NPWT) or skin grafting.

Wounds were categorized by etiology (venous, diabetic, arterial, post-traumatic) and location (lower limb or other). Microbiological cultures were obtained when infection was suspected.

When infection was clinically suspected on the basis of the wound’s appearance (e.g., purulence, odor, erythema, or delayed healing), wound swabs were obtained under sterile conditions. Specimens were collected and processed using standard aerobic and anaerobic culture methods. Bacterial identification and antibiotic susceptibility testing were performed in accordance with EUCAST guidelines. The presence of multi-drug resistant (MDR) organisms was defined according to international criteria. Platelet-rich plasma (PRP) was prepared using 10 mL Newlife^®^ ACDA tubes, which contain a high concentration of sodium citrate as an anticoagulant. These tubes, equipped with a separation gel, allow efficient partitioning of blood components and preservation of essential bioactive molecules. Blood samples were centrifuged at 3500 revolutions per minute (RPM) for 10 min, yielding a high-purity PRP fraction. From each tube, approximately 4–5 mL of PRP was obtained. The final product was administered under sterile conditions by infiltration at an intradermal level at the wound margins and at a depth of 2–3 mm beneath the wound to ensure local delivery directly to the wound microenvironment.

Autologous fat grafting was performed under local anesthesia using Klein solution infiltration at the donor site, typically the inner thigh or abdominal region. Fat was harvested using a 3 mm Mercedes-tip cannula through gentle manual aspiration. The lipoaspirate was then processed using a standardized protocol that included decantation, repeated washing with saline, and mechanical emulsification to reduce particle size. Emulsified fat was passed through 1.2 mm filters to obtain micro-fat, ensuring a more uniform and injectable consistency. The refined graft was subsequently injected into the wound bed and perilesional tissues using a 19-gauge blunt-tip facial fat grafting cannula.

Following surgical debridement, all patients received at least one session of platelet-rich plasma (PRP) therapy, administered via perilesional infiltration under sterile conditions. In selected cases—primarily those presenting with larger or refractory wounds and a suitable donor site—autologous fat grafting was also performed as an adjunctive treatment. The decision to use fat grafting was based on clinical assessment of the wound’s chronicity, depth, and tissue viability and the patient’s general condition. When both therapies were indicated, PRP and fat grafts were applied during the same operative session.

In patients requiring surgical coverage, split-thickness skin grafts (STSGs) were harvested using a Zimmer^®^ electric dermatome, set to an intermediate thickness of approximately 0.30 mm. The harvested grafts were applied to the wound bed and secured with metallic skin staples. Graft sites were dressed with Vaseline-impregnated gauze combined with chlorhexidine mesh and left undisturbed for approximately five days postoperatively. The donor sites were covered with Vaseline gauze alone. The first dressing change was typically performed on postoperative Day 5, with subsequent wound care based on clinical assessments of graft adherence and healing progress.

All patients received standardized local wound care and regular dressing changes in accordance with institutional protocols.

### 2.3. Statistical Analysis

Data analysis was performed using IBM SPSS Statistics v.25, and graphical illustrations were created with Microsoft Excel/Word 2024. Quantitative variables were assessed for normal distribution using the Shapiro–Wilk test. The results were expressed as the mean ± standard deviation (SD) or median and interquartile range (IQR), as appropriate.

Group comparisons were conducted using the following:Student’s T-test for normally distributed quantitative variables;Mann–Whitney U test for non-parametric data;Fisher’s exact test for categorical variables.

Statistical significance was defined as *p* < 0.05.

## 3. Results

### 3.1. Patient Demographics and Clinical Characteristics

The study included 35 patients, of whom 54.3% (n = 19) were men. The mean age was 59.11 ± 13.64 years, with no statistically significant difference between patients who received skin grafts and those who did not (*p* = 0.737). A significant difference was observed in BMI: patients with skin grafts had a higher average BMI (31.41 ± 7.22) compared with those without grafts (25.69 ± 3.42; *p* = 0.009). Regarding smoking status, 45.7% of patients were smokers, and those without skin grafts had a significantly higher prevalence of smoking (71.4%) than those with grafts (28.6%; *p* = 0.018) (Table 1, Figure 1 and Figure 2).

### 3.2. Wound Etiology, Localization, and Duration

The most common wound etiologies were venous (40.0%), diabetic (28.6%), post-traumatic (20.0%), and arterial ulcers (11.4%). The majority of ulcers (91.4%) were located on a lower extremity. Wound surface area was significantly larger in the skin graft group (median = 68 cm^2^) compared with the no graft group (median = 19 cm^2^; *p* = 0.002). Similarly, wounds were older in the graft group (median = 24 weeks) than in the non-graft group (median = 7.5 weeks; *p* = 0.024) (Table 2, Figure 3 and Figure 4).

### 3.3. Association Between Clinical Variables and Use of Skin Grafts

Clinical comorbidities included diabetes (40.0%), chronic venous insufficiency (48.6%), peripheral artery disease (20.0%), and autoimmune/vasculitis-related conditions (20.0%). Although none of these differences reached statistical significance, chronic venous insufficiency was more prevalent in the graft group (61.9%) than in the non-graft group (28.6%) (Table 3).

### 3.4. Treatment Modalities and Therapeutic Interventions

Therapeutic approaches included negative pressure wound therapy (NPWT), platelet-rich plasma (PRP) sessions, and autologous fat tissue interventions. NPWT was used in 22.9% of patients, with a higher proportion in the graft group (33.3% vs. 7.1%). Most patients received only one PRP session (82.9%). Treatment duration showed no statistically significant difference between groups (Table 4).

### 3.5. Healing Outcomes and Wound Surface Reduction

Median healing progress was 90%, with no significant difference between groups (*p* = 0.606). Complete healing was achieved in 45.7% of patients, while 34.3% showed no improvement. The proportions of complete healing were nearly equal in both groups (42.9% vs. 47.6%) (Table 5).

### 3.6. Infection Profile and Microbiological Findings

Microbiological analysis revealed that *Pseudomonas aeruginosa* was the most frequently isolated pathogen (37.1%), and its presence was significantly more common in patients with skin grafts (52.4%) compared with those without (14.3%; *p*= 0.034). Other bacteria included *S. aureus*, *E. coli*, and multi-drug resistant strains, but no statistically significant differences were found between groups for these organisms (Table 6, Figure 5).

### 3.7. Inflammatory Markers and Laboratory Parameters

Inflammatory markers such as the neutrophil-to-lymphocyte ratio (NLR) and fibrinogen levels did not differ significantly between groups. The median NLR was 2, and fibrinogen levels averaged 405 mg/dL across both groups (Table 7).

### 3.8. Use of Antibiotic Therapy and Duration

A significantly higher proportion of patients with skin grafts received antibiotic treatment (95.2%) compared with those without grafts (42.9%; *p* = 0.001). However, the duration of antibiotherapy was similar in both groups, with a median of 7 days (Table 8, Figure 6).

## 4. Discussion

The findings of this study confirm the effectiveness of regenerative interventions, namely platelet-rich plasma (PRP) and autologous fat grafting, in the management of chronic wounds, offering valuable insights into their practical application. The statistical analysis revealed associations between demographic characteristics, comorbidities, etiologies, inflammatory markers, and treatment outcomes, allowing for a nuanced and evidence-based interpretation.

The mean age of 59 years and male predominance in our cohort are demographic features commonly reported in studies involving patients with chronic wounds [23]. The significantly higher BMI observed in the graft group (31.4 vs. 25.7) highlights the impact of obesity on wound severity. This association is supported by the existing literature, which identifies obesity as a driver of systemic chronic inflammation, impaired local perfusion, and disruption of the wound healing cascade [24,25]. One study found that obesity is linked to an increased risk of venous and arterial ulcers, prolonged treatment needs, and greater postoperative complication rates [26], while another underlines the negative impact of adipose tissue on mesenchymal stromal cells, thereby reducing the skin’s regenerative capacity [27].

With respect to smoking, our findings revealed a significantly higher prevalence in the non-graft group (71.4% vs. 28.6%), which may tacitly suggest lower surgical eligibility or delayed access to operative interventions. Smoking is well documented to delay wound healing through mechanisms such as vasoconstriction, tissue hypoxia, fibroblast dysfunction, and reduced collagen synthesis [28,29]. Data from the American College of Surgeons National Surgical Quality Improvement Program (ACS NSQIP) indicate a 65% higher risk of wound dehiscence and increased infection rates among smokers [30]. Consequently, institutions such as the World Health Organization recommend a minimum four-week period of abstinence from smoking prior to surgery to mitigate perioperative risks [31]. These correlations emphasize that obesity and smoking are critical determinants in the prognosis of chronic wounds, rather than mere demographic variables. In clinical practice (as seen in Figure 7 and Figure 8), the success of regenerative treatments, such as PRP and fat grafting is closely linked to the management of systemic metabolism and environmental factors.

Glycemic control and weight reduction may optimize perfusion and inflammatory responses, while structured smoking cessation programs reduce the risk of infection and delayed healing. Integrated interventions combining nutritional counselling, metabolic and psychological support, and cessation assistance represent the most effective approach to both the prevention and treatment of chronic wounds.

Most wounds were located on the lower limbs (91.4%), with venous (40%) and diabetic (28.6%) etiologies being the most prevalent. These findings are consistent with patterns widely described in the literature [32]. The incidence of venous leg ulcers, which contribute significantly to the global burden of chronic wounds, is estimated at up to 1% among adults, particularly older individuals. These ulcers are often persistent, recurrent, and challenging to heal [32,33]. The markedly larger median size of wounds in the graft group (68 cm^2^ vs. 19 cm^2^; *p* = 0.002), alongside their greater duration (24 vs. 7.5 weeks; *p* = 0.024), suggests a more advanced state of chronicity and a biologically entrenched barrier to healing. These are commonly associated with the need for complex interventions such as skin grafting (as seen in Figure 9 and Figure 10) or advanced biological therapies [34].

Clinical guidelines recognize wound size and duration as key prognostic indicators of healing. Ulcers that fail to demonstrate meaningful improvement within four weeks of standard care are considered refractory and may warrant additional treatment strategies [35,36]. Furthermore, chronic venous ulcers are characterized by a persistent pro-inflammatory tissue microenvironment and dysregulated angiogenesis. These are marked by leucocyte infiltration, protease activity, and oxidative stress, collectively creating a toxic biochemical profile that impairs tissue regeneration [37].

In diabetic ulcers, fibroblasts exhibit markedly diminished migratory capacity and extracellular matrix synthesis, accompanied by hyperactivation of matrix metalloproteinases (MMP-2, MMP-9), up to 30 times more intense than in acutely healing tissue, resulting in excessive matrix degradation and an inability to transition from the inflammatory to the proliferative phase [38].

The necessity for regenerative interventions is further evidenced by the fact that large wounds (>25 cm^2^) of prolonged duration are the most frequent candidates for partial thickness skin grafts; these grafts promote healing by coverage, exudate reduction, and activation of local reparative mechanisms [39].

Thus, the alignment of our data, the patients’ clinical characteristics, and current scientific evidence highlights the importance of employing targeted approaches, including systematic evaluation of wound size and duration to determine the optimal timing and technology for intervention.

Negative pressure wound therapy (NPWT) was used significantly more often in the graft group (33.3% vs. 7.1%, *p* = n.s.), suggesting that this adjunctive method is selected for more complex wounds. The literature indicates that NPWT accelerates healing and reduces infection rates in chronic and post-traumatic wounds [36,37]. In a randomized trial comparing NPWT with its instillation variant (NPWTi-d), the median time to wound closure decreased from 10 to 6 days (*p* < 0.001), demonstrating increased efficacy in treating complex lesions [36]. Moreover, a recent meta-analysis reported a significant reduction in infections (RR = 0.59; *p* < 0.001) compared with standard care [37].

Regarding platelet-rich plasma (PRP) therapy, it was administered predominantly in a single session (82.9%), while only 17.1% of patients received two or more sessions. Evidence in the literature supports the effectiveness of PRP in accelerating healing and reducing the size of chronic wounds. One study reported a significant increase in complete healing rates (RR = 1.20; 95% CI: 1.09–1.32), alongside a mean reduction in healing time of 4 to 40 days [39]. However, the quality of evidence is considered moderate to low, underscoring the need for robust, well-designed clinical trials [38].

Fat grafting was employed in 17.1% of cases, particularly selected for extensive wounds that were resistant to conventional treatment. The integration of autologous adipocytes, rich in adipose-derived stem cells (ADSCs), stimulates angiogenesis and regenerates the extracellular matrix [39]. Scientific reviews have confirmed that adipocytes isolated from fat, when combined with PRP, enhance graft viability and microvascular density, demonstrating promising synergistic effects [40,41].

The absence of a significant difference in the median treatment duration (~35 days) between the groups indicates that incorporating advanced therapies such as NPWT, PRP, and autologous fat grafting does not prolong overall healing time. This suggests comparable effectiveness across treatment modalities, even when complex protocols are employed.

The integration of NPWT in cases requiring skin grafting clearly offers benefits in exudate management, promotion of granulation, and infection prevention, as supported by robust evidence from randomized controlled trials and meta-analyses [36,37,38]. PRP remains an effective option for enhancing closure rates and accelerating healing, even with a limited number of sessions, though higher-quality studies are needed to clarify its optimal role [39]. Autologous fat grafting, particularly when combined with PRP, shows promising potential through activation of tissue regeneration, as emerging preclinical and clinical data indicate [40,41].

These findings endorse a multimodal approach combining NPWT, PRP, and fat grafts in the management of complex chronic wounds, emphasizing maximization of biological synergies and minimization of therapeutic risks.

An average healing rate of 90% and complete healing in 45.7% of the cohort reflect real and encouraging clinical efficacy. The lack of statistically significant differences between grafted and non-grafted patients further highlights the potential of regenerative therapies: by employing PRP and autologous fat grafting, comparable outcomes were achieved even in complex lesions. This supports the hypothesis that regenerative interventions can level healing prognoses between moderate and severe wounds without compromising efficacy.

Benchmark studies support these conclusions: a 2022 meta-analysis reported a significant increase in complete healing rates for PRP-treated wounds compared with the controls (RR = 1.25; 95% CI: 1.10–1.42), with an average reduction in healing time of approximately 26 days [42]. Additionally, the inclusion of ADSCs via autologous fat grafting has been associated with a higher capillary density and reduced fibrotic scarring, correlating with more durable healing [43].

A clinical randomized trial demonstrated that diabetic wounds treated with PRP plus fat grafting achieved complete healing in 58% of cases compared with 38% in the standard care group (*p* = 0.03) [44], indicating that combined therapies may outperform single interventions.

Nevertheless, caution is warranted: the quality and frequency of regenerative treatments vary across studies, and standardized protocols concerning PRP concentration, graft volumes, and application intervals are needed.

In the context of our data, average healing of 90% and complete healing in 45.7% across both moderate and severe wounds were found; these findings support the important role of integrated regenerative therapies. They may offer equivalent efficacy to grafting while reducing the complexity of surgical procedures for severe lesions. Clinically, this provides strong rationale for developing therapeutic protocols that include PRP and autologous fat grafting in the management of severe chronic wounds, with the goals of improving healing rates, reducing recurrence, and minimizing the need for reoperation.

Colonization with *Pseudomonas aeruginosa* was significantly more frequent in the skin graft group (52.4% vs. 14.3%, *p* = 0.034), suggesting its presence reflects the initial severity and advanced chronicity of the wound rather than being a consequence of treatment [45]. This pattern is well documented: *P. aeruginosa* forms resilient biofilms that penetrate deep into wounds and impede tissue regeneration [46,47]. Studies show that its presence correlates with larger wounds, delayed healing, and higher graft failure rates [48].

The biofilms produced by this bacterium act as barriers against both antibiotics and the immune response, sustaining chronic inflammation [49]. Systematic microbiological screening and sensitivity testing allow for targeted management of colonization. Clinically, wound swabs or biopsies followed by culture and antibiotic susceptibility testing are recommended for any wound showing signs of infection or suspected colonization [50].

Antimicrobial treatment, both local (e.g., acetic acid >1%) and systemic, should be guided by culture results to ensure effective action against *P. aeruginosa*, thereby avoiding under-dosing or indiscriminate antibiotic use that may promote resistance [51]. Current guidelines advocate combined approaches (debridement plus topical or systemic antibiotics) for wounds colonized by *P. aeruginosa*, accompanied by continuous monitoring and repeated microbiological assessment [52].

Consequently, *P. aeruginosa* colonization represents more than a passive finding; it signals the need for proactive intervention. Systematic screening and appropriate treatment can shorten healing time, prevent graft failure, and reduce the risk of systemic infections such as osteomyelitis.

Mean values of neutrophil-to-lymphocyte ratio (NLR) and fibrinogen did not differ significantly between the graft and non-graft groups (median NLR ≈ 2 vs. 3, fibrinogen ≈ 405 mg/dL; *p* > 0.05). This implies that the regenerative therapies PRP and autologous fat grafting did not provoke excessive systemic inflammation. These findings support the recognized anti-inflammatory safety profile of such interventions [51].

Meta-analyses and Cochrane reviews on PRP use in chronic wounds report no significant increase in infections or adverse events compared with conventional treatments [51]. For instance, Mayo Clinic data affirm that PRP presents a neutral risk concerning infection and systemic complications [53,54].

Molecular mechanisms indicate that PRP acts via controlled local release of cytokines and growth factors, modulating the inflammatory response and promoting tissue regeneration while maintaining systemic homeostasis without triggering acute inflammation [55,56].

Similarly, autologous fat grafting, rich in adipose-derived stromal cells (ADSCs), exhibits notable immunomodulatory effects. ADSCs encourage macrophages to switch to the reparative M2 phenotype, thereby reducing local inflammation and supporting extracellular matrix regeneration [57,58]. Recent reviews have indicated that these cells, through paracrine secretion and cellular interactions, regulate immune responses and promote healing without causing systemic inflammatory disturbances [59,60].

The absence of systemic changes in NLR and fibrinogen within our cohort thus supports both theoretical models and existing clinical data: PRP and autologous fat grafting can profoundly modulate local inflammation while preserving overall immune homeostasis. These findings reinforce the safety profile of regenerative therapies in chronic wound management, supporting their integration into clinical protocols to facilitate efficient healing without systemic adverse effects.

Antibiotic therapy was required in 95.2% of skin graft cases, compared with 42.9% in the non-graft group (*p* = 0.001), while the median duration of seven days was similar across groups (IQR 7–7.5 vs. 7–8.5 days). These results indicate that while complex chronic wounds often necessitate antibiotic use, such intervention does not prolong exposure unnecessarily, aligning with antibiotic stewardship principles.

Specialized literature demonstrates that in cutaneous surgery, pre- and perioperative antibiotic prophylaxis increases treatment frequency but does not significantly improve graft survival or infection rates, especially for split-thickness skin grafts [61,62]. Studies concerning chronic wounds recommend systemic antibiotics only for clinically proven infections, typically over approximately 10 days, with extended regimens (over 6 weeks) indicated for osteomyelitis [7].

The IDSA guidelines for non-purulent skin infections recommend brief, five-day antibiotic courses, extending the duration only if clinical improvement is insufficient [63]. In reconstructive surgery, prophylactic antibiotics are avoided in the absence of risk factors (e.g., implants, immunosuppression), favoring local antiseptics or non-antibiotic strategies [64,65].

For chronic wounds requiring grafts, a one-week empirical systemic antibiotic course is justified, provided that reevaluation occurs, based on antibiograms to prevent overuse and resistance development. Our observed seven-day duration aligns with these recommendations and does not unduly prolong antibiotic exposure.

Thus, the higher antibiotic use in graft recipients reflects the need for a preventative and therapeutic protocol tailored to complex clinical profiles. However, the limited one-week duration strikes a balance between efficacy and bacterial resistance prevention.

The favorable outcomes of this study support the integration of PRP and fat grafting into treatment protocols for chronic wounds. The cellular mechanisms involved are multifaceted: PRP enhances angiogenesis (via VEGF), fibroblast proliferation (EGF, FGF), recruitment of reparative cells (PDGF, IGF), and modulation of the inflammatory response (TGF-β) and provides antibacterial potential through microbicidal proteins all within a safe, autologous framework. Fat grafting contributes to regeneration through stromal cells with multipotent capacity, antioxidant properties, modulation of local macrophages, and extracellular matrix remodeling.

Previous research has demonstrated the consistent benefits of PRP in diabetic, pressure, and venous ulcers, notably accelerating healing and granulation tissue formation. Preliminary data on autologous fat grafting indicate good graft survival in well-vascularized tissues; however, retention rates may decline in poorly vascularized wound environments. Our findings suggest that combining PRP with fat grafting may enhance therapeutic outcomes. Nonetheless, confirmation requires randomized controlled trials featuring separate treatment arms and standardized protocols.

This study has several limitations: a relatively small sample size (n = 35), an observational design, absence of a non-regenerative control group, and a relatively short follow-up period. One key limitation of this study is the relatively small sample size and the unequal distribution between the regenerative-only group and the graft-assisted group. This imbalance resulted from real-world clinical decision-making based on wound characteristics, patients’ comorbidities, and surgical eligibility, rather than randomized allocation. As a result, the statistical power to detect between-group differences may be reduced, and the generalizability of the findings to broader populations may be limited. Future studies with larger cohorts and randomized controlled designs are necessary to validate these results, minimize selection bias, and allow for more robust subgroup analyses. Additionally, the heterogeneity in PRP preparation (platelet concentration, activation methods) and in fat grafting techniques reflected real-world practices but limited reproducibility. Future studies should incorporate quantitative analyses of local and systemic biological mediators (e.g., serial VEGF, PDGF, TGF-β levels, and NLR), detailed assessments of fat graft quality (retention rates, microvascular viability), and cost–benefit evaluations compared with standard care and other advanced therapies.

## 5. Conclusions

Regenerative interventions, including PRP and autologous fat grafting, have proven effective in promoting the healing of chronic wounds, achieving comparable healing rates even in severely affected lesions without inducing significant systemic inflammatory responses. These regenerative approaches demonstrate a favorable safety profile and are suitable for integration into multidisciplinary treatment protocols. We recommend conducting randomized controlled trials and developing standardized protocols to validate and optimize their use in chronic wound management.

## Figures and Tables

**Figure 1 biomedicines-13-01827-f001:**
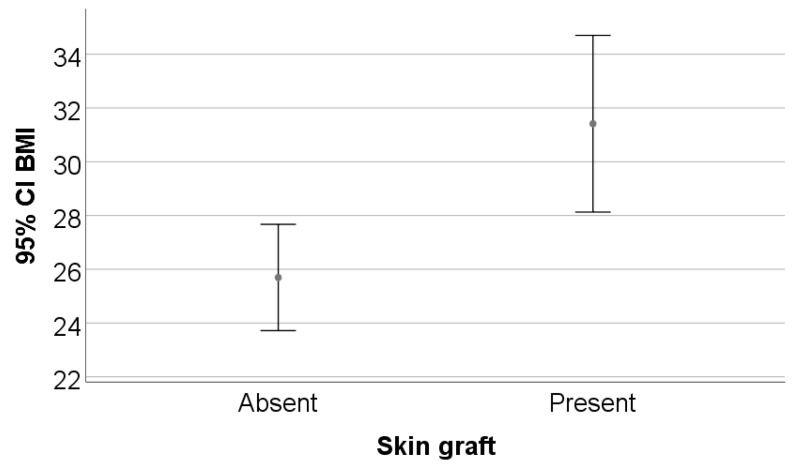
BMI comparison between grafted and non-grafted groups with error representation. Data are presented as the mean ± standard deviation (SD), consistent with Table 1.

**Figure 2 biomedicines-13-01827-f002:**
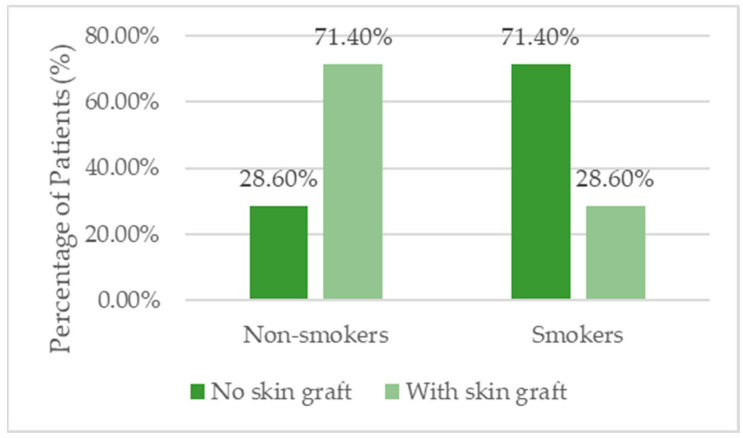
Distribution of smoking status among patients treated with skin grafting (graft-assisted group) versus those receiving regenerative therapy alone (regenerative-only group).

**Figure 3 biomedicines-13-01827-f003:**
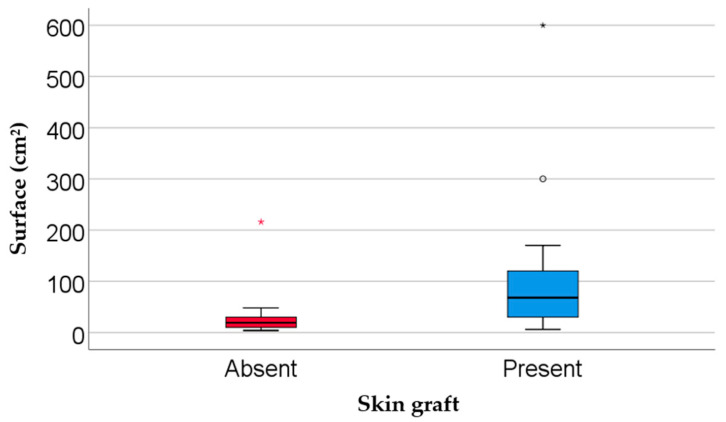
Comparison of wound surface area by skin graft status. Data are presented as the median (interquartile range, * represents the values as extreme outliers); statistical significance: *p* = 0.002.

**Figure 4 biomedicines-13-01827-f004:**
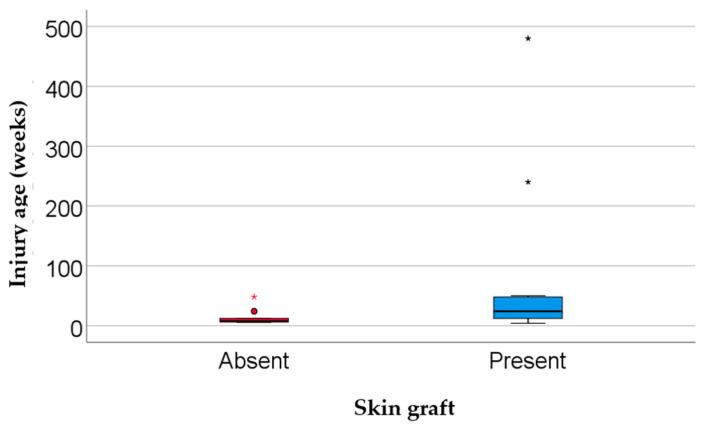
Comparison of wound duration by skin graft status. Data are presented as the median (interquartile range, * represents the values as extreme outliers); statistical significance: *p* = 0.024.

**Figure 5 biomedicines-13-01827-f005:**
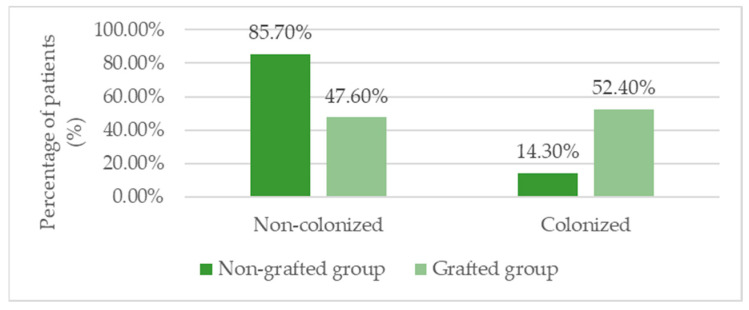
Prevalence of *Pseudomonas aeruginosa* colonization in grafted vs. non-grafted groups.

**Figure 6 biomedicines-13-01827-f006:**
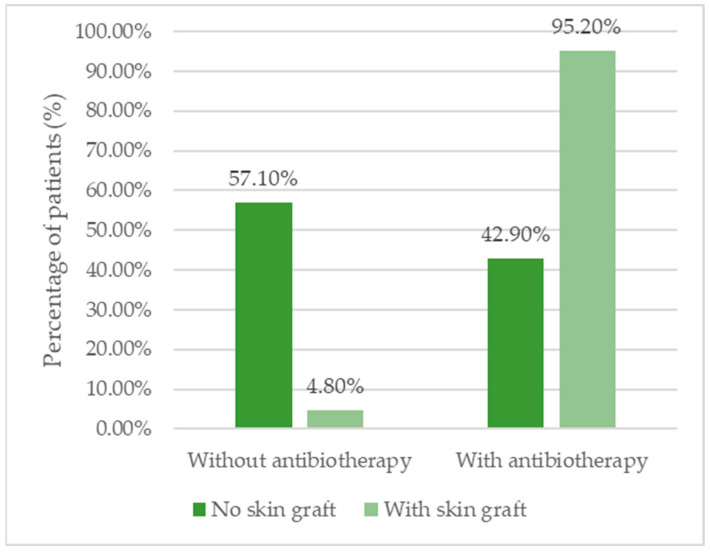
Distribution of the patients according to the existence of skin grafts and necessity for antibiotherapy.

**Figure 7 biomedicines-13-01827-f007:**
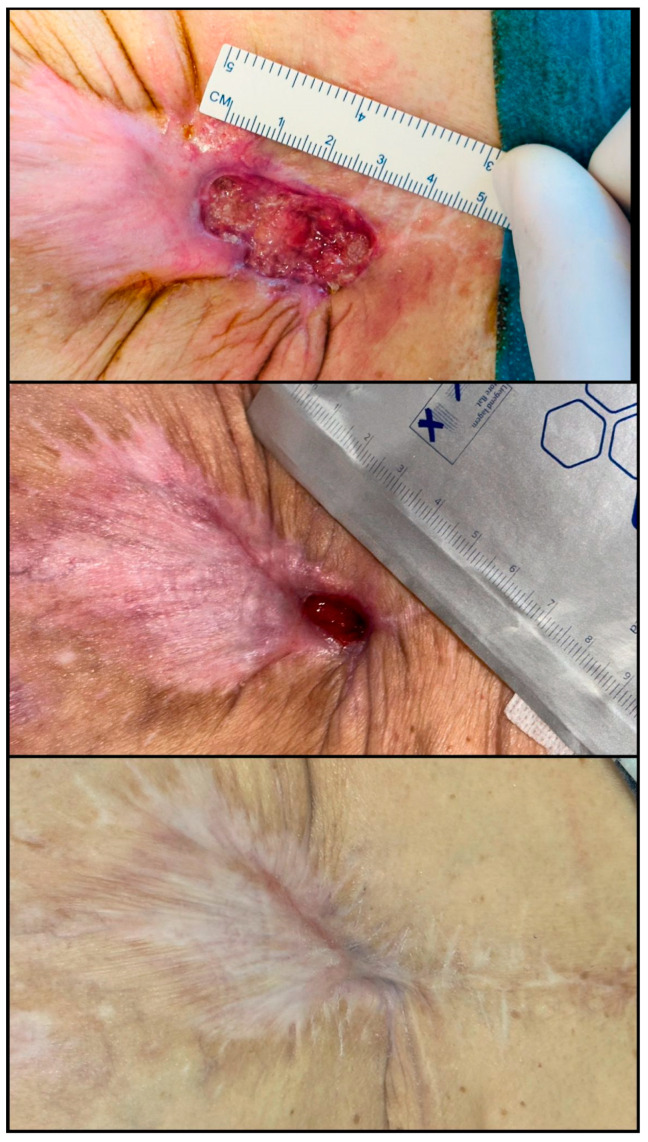
Chronic postoperative abdominal wound secondary to the rejection of a non-resorbable surgical mesh, colonized by Pseudomonas aeruginosa. The lesion had persisted for approximately one year with minimal improvement despite conservative management. Regenerative therapy included one session of platelet-rich plasma (PRP) combined with autologous fat grafting, followed by a second PRP session administered 14 days later. The wound initially measured approximately 6 × 3 cm. Slow granulation and epithelialization were noted within 6 weeks, with over 80% area reduction by Day 60 and complete wound closure achieved by Day 90 (original clinical image from the authors’ case archive).

**Figure 8 biomedicines-13-01827-f008:**
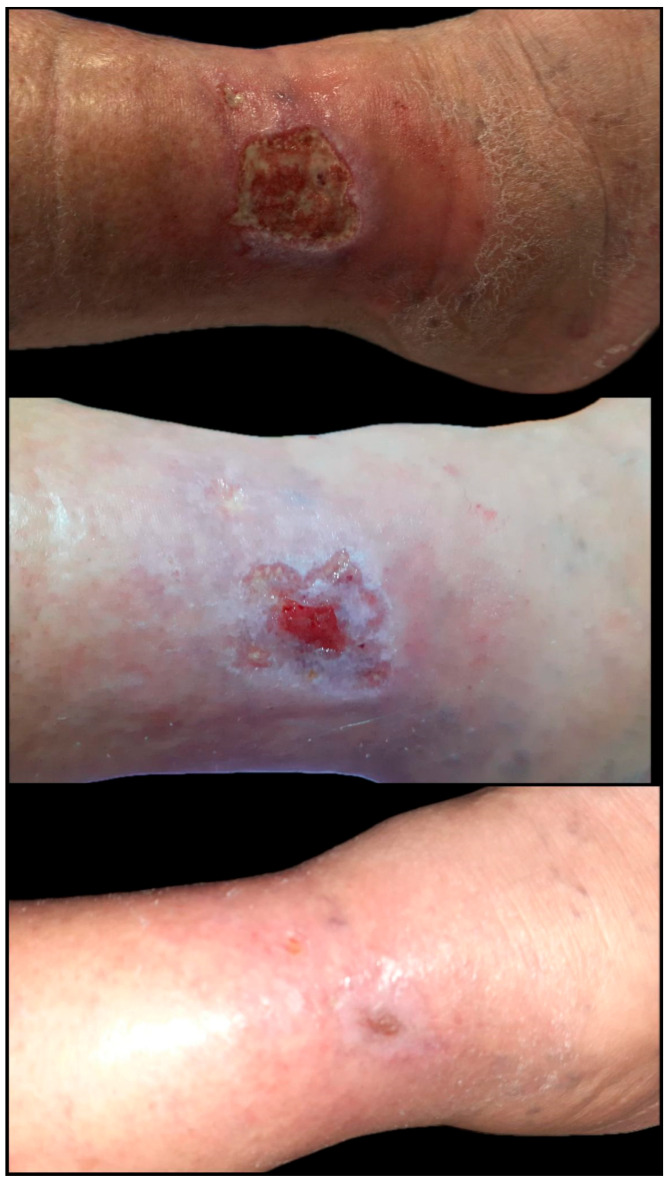
Chronic diabetic ulcer located on the lower leg, unresponsive to conservative management over a 60-day period. The wound measured approximately 5 × 4 cm at initial assessment and showed no signs of granulation or epithelial advancement. Following surgical debridement, a single session of platelet-rich plasma (PRP) therapy was administered via perilesional and intralesional infiltration. Rapid improvement was observed, with over 75% wound area reduction by Day 21 and complete epithelialization achieved within four weeks post-intervention (original clinical image from the authors’ case archive).

**Figure 9 biomedicines-13-01827-f009:**
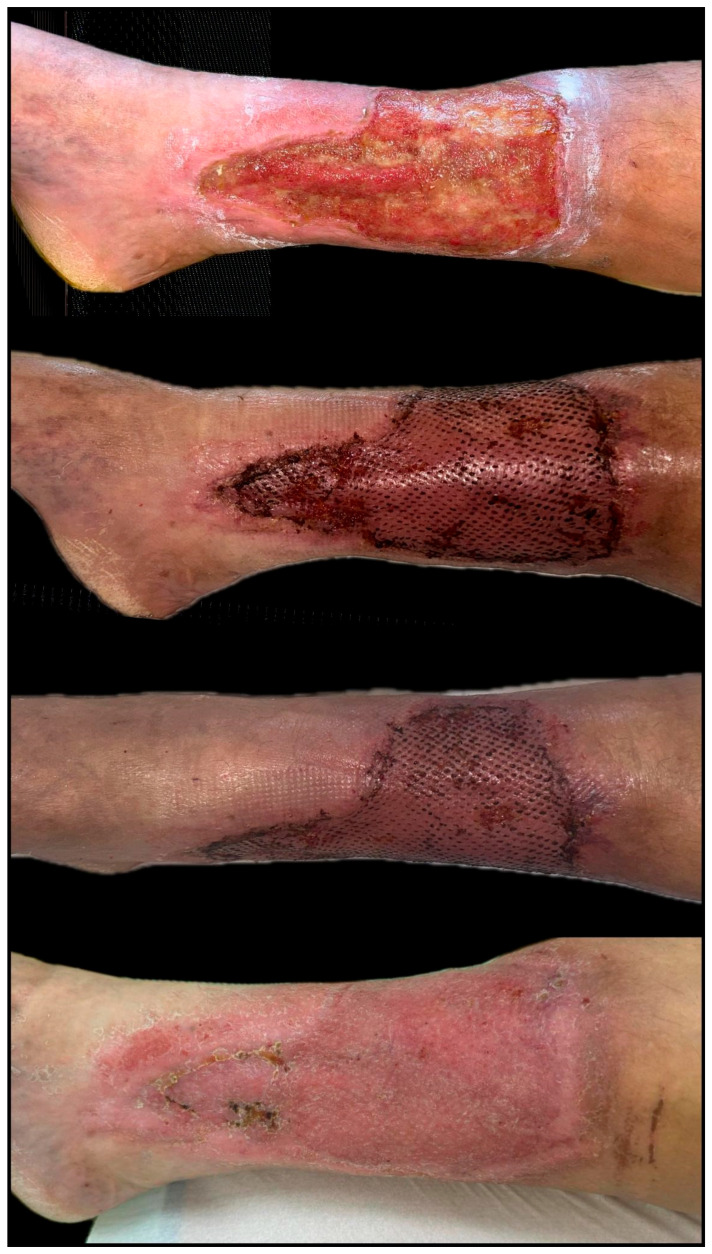
Extensive chronic venous ulcer of the lower leg, measuring 20 × 12 cm, with a two-year history of non-healing despite repeated conservative therapy, multiple surgical debridements, and two failed split-thickness skin grafting attempts. The patient underwent a single session of autologous fat grafting combined with platelet-rich plasma (PRP) therapy, followed by the application of a split-thickness skin graft. A second PRP session was administered 14 days later. Significant granulation and graft adherence were observed within three weeks, with > 85% wound area closure by Day 30. Complete epithelialization and wound healing were achieved within six weeks post-intervention (original clinical image from the authors’ case archive).

**Figure 10 biomedicines-13-01827-f010:**
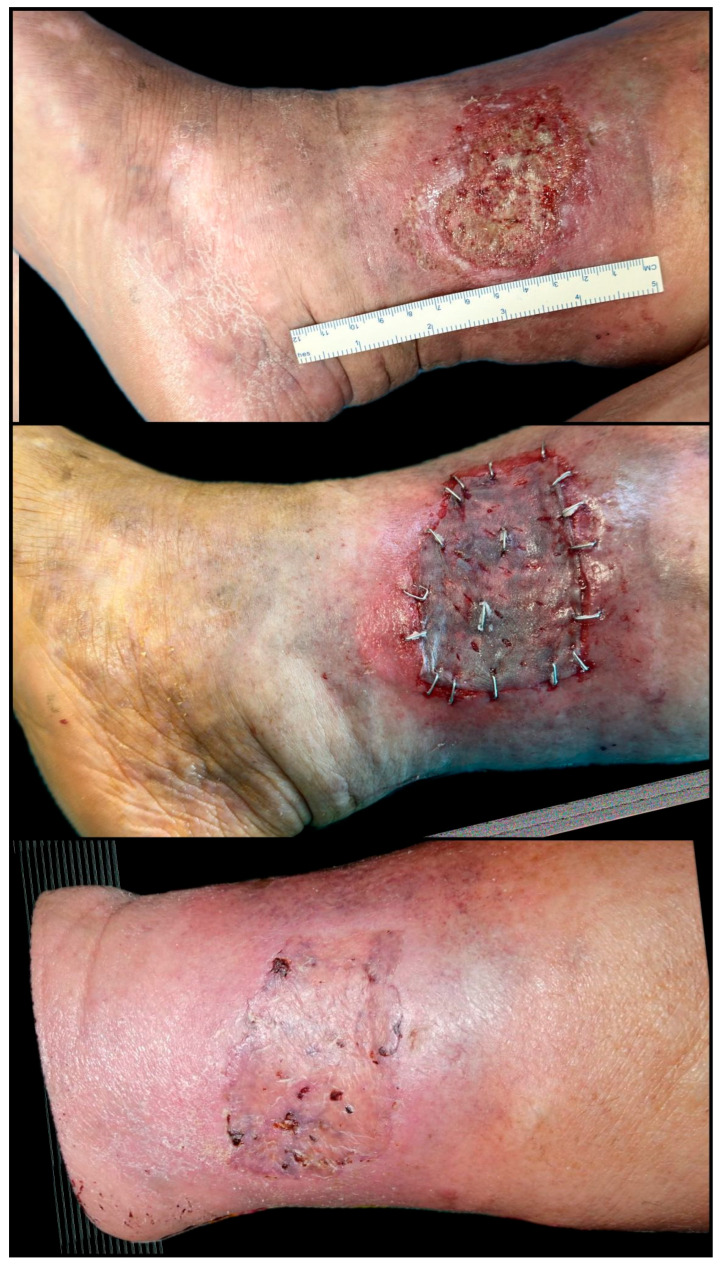
Chronic inflammatory and autoimmune ulcer of the lower leg, with a one-year history of poor evolution and resistance to conservative management. The ulcer measured approximately 10 × 7 cm at the time of intervention. The patient received one session of platelet-rich plasma (PRP) therapy combined with a split-thickness skin graft. Rapid improvement was observed, with excellent graft take and >90% wound closure by Day 14. Complete epithelialization and full healing were achieved within 4 weeks post-procedure (original clinical image from the authors’ case archive).

**Table 1 biomedicines-13-01827-t001:** Demographic characteristics of the patients (* Student’s *t*-test; ** Fisher’s exact test).

Parameter	Total (n = 35)	Regenerative-Only Group (n = 14)	Graft-Assisted Group (n = 21)	*p*-Value
Age (mean ± SD)	59.11 ± 13.64	58.14 ± 14.20	59.76 ± 13.58	0.737 *
Gender (male, n/%)	19 (54.3%)	9 (64.3%)	10 (47.6%)	0.491 **
BMI (mean ± SD)	29.12 ± 6.57	25.69 ± 3.42	31.41 ± 7.22	0.009 *
Smoking (n/%)	16 (45.7%)	10 (71.4%)	6 (28.6%)	0.018 **

**Table 2 biomedicines-13-01827-t002:** Summary of wound etiology, anatomical location, surface area, and duration, stratified by grafting status. (** Fisher’s exact Test; *** Mann–Whitney U test).

Parameter	Total	Regenerative-Only Group	Graft-Assisted Group	*p*-Value
Venous ulcers (n/%)	14 (40%)	3 (21.4%)	11 (52.4%)	0.194 **
Arterial ulcers (n/%)	4 (11.4%)	3 (21.4%)	1 (4.8%)	
Diabetic ulcers (n/%)	10 (28.6%)	5 (35.7%)	5 (23.8%)	
Post-traumatic ulcers (n/%)	7 (20%)	3 (21.4%)	4 (19%)	
Localization—lower extremity (n/%)	32 (91.4%)	13 (92.9%)	19 (90.5%)	1.000 **
Surface area (cm^2^, median IQR)	45 (18–110)	19 (9.75–34.5)	68 (27.5–123)	0.002 ***
Injury age (weeks, median IQR)	12 (6–48)	7.5 (6–15)	24 (9.5–49)	0.024 ***

**Table 3 biomedicines-13-01827-t003:** Associations of comorbidities, clinical factors, and skin graft use; ** Fisher’s exact test.

Parameter	Total	Regenerative-Only Group	Graft-Assisted Group	*p*-Value
Diabetes (n/%)	14 (40%)	5 (35.7%)	9 (42.9%)	0.737 **
Chronic venous insufficiency (n/%)	17 (48.6%)	4 (28.6%)	13 (61.9%)	0.086 **
Peripheral artery disease (n/%)	7 (20%)	4 (28.6%)	3 (14.3%)	0.401 **
Autoimmune/vasculitis (n/%)	7 (20%)	1 (7.1%)	6 (28.6%)	0.203 **

**Table 4 biomedicines-13-01827-t004:** Use of adjunctive therapies and procedural data (** Fisher’s exact test; *** Mann–Whitney U test).

Parameter	Total	Regenerative-Only Group	Graft-Assisted Group	*p*-Value
NPWT (n/%)	8 (22.9%)	1 (7.1%)	7 (33.3%)	0.108 **
PRP-1 session (n/%)	29 (82.9%)	13 (92.9%)	16 (76.2%)	0.366 **
PRP-≥2 sessions (n/%)	6 (17.1%)	1 (7.1%)	5 (23.8%)	
Fat tissue intervention (n/%)	6 (17.1%)	2 (14.3%)	4 (19%)	1.000 **
Treatment duration (days, IQR)	35 (25–70)	39 (17–67.75)	34 (28–165)	0.752 ***

**Table 5 biomedicines-13-01827-t005:** Healing progression outcomes in grafted and non-grafted groups (** Fisher’s exact test; *** Mann–Whitney U test).

Parameter	Total	Regenerative-Only Group	Graft-Assisted Group	*p*-Value
Healing progress (%) (IQR)	90 (10–100)	87.5 (0–100)	90 (40–100)	0.606 ***
No improvement (n/%)	12 (34.3%)	6 (42.9%)	6 (28.6%)	0.737 **
Satisfactory results (n/%)	7 (20%)	2 (14.3%)	5 (23.8%)	
Complete healing (n/%)	16 (45.7%)	6 (42.9%)	10 (47.6%)	

**Table 6 biomedicines-13-01827-t006:** Distribution of bacterial isolates by skin graft status (** Fisher’s exact test).

Parameter	Total	Regenerative-Only Group	Graft-Assisted Group	*p*-Value
*P. aeruginosa* (n/%)	13 (37.1%)	2 (14.3%)	11 (52.4%)	0.034 **
*S. aureus* (n/%)	6 (17.1%)	1 (7.1%)	5 (23.8%)	0.366 **
*E. coli* (n/%)	7 (20%)	3 (21.4%)	4 (19%)	1.000 **
Other bacteria (n/%)	8 (22.9%)	1 (7.1%)	7 (33.3%)	0.108 **
MDR bacteria (n/%)	12 (34.3%)	3 (21.4%)	9 (42.9%)	0.282 **

**Table 7 biomedicines-13-01827-t007:** Laboratory and inflammatory parameters (*** Mann–Whitney U test).

Parameter	Total	Regenerative-Only Group	Graft-Assisted Group	*p*-Value
NLR (median, IQR)	2 (2–4)	2 (1.75–4.25)	3 (2–4)	0.778 ***
Fibrinogen (median, IQR)	405 (346–478)	398 (345–484)	405 (344–473)	0.829 ***

**Table 8 biomedicines-13-01827-t008:** Use and duration of antibiotic therapy (*** Mann–Whitney U test).

Parameter	Total (n = 35)	Regenerative-Only Group (n = 14)	Graft-Assisted Group (n = 21)	*p*-Value
Antibiotherapy administration rate (n/%)	26 (74.3%)	6 (42.9%)	20 (95.2%)	0.001
Duration of antibiotherapy (days, IQR)	7 (7–7.5)	7 (7–8.75)	7 (7–8.5)	1.000 ***

## Data Availability

Data supporting the findings of this study are not publicly available due to privacy and ethical restrictions involving patient confidentiality. Access to anonymized data may be granted upon reasonable request and with appropriate institutional approval.

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
