# Peer review of "Integrating Regenerative Medicine in Chronic Wound Management: A Single-Center Experience"

_biomedicines, 2025, doi:10.3390/biomedicines13081827_

Round 1
Reviewer 1 Report
Comments and Suggestions for Authors
Manuscript Evaluation Report
Manuscript Title: “Integrating Regenerative Medicine in Chronic Wound Management: A Single-Center Experience”
Journal: Biomedicines
Section: Molecular and Translational Medicine
I. General Observation:
The manuscript addresses an important clinical issue in wound care by using PRP and autologous fat grafting, which are regenerative therapies. The study provides an observational analysis of treatment outcomes and supports their integration into chronic wound protocols.
1. Major Comments:
1.1 Small sample size and unequal grouping
The study involves only 35 patients, with 14 patients in the regenerative group and 21 in the graft-assisted group. Group imbalance can affect statistical power.
1.2 Image legends lack scientific detail:
Figure descriptions (figures 7-10) do not specify healing, time intervals, or quantitative assessment.
2. Minor comments:
Enhance the figures by adding clinical data in the legends.
3. Recommendation:
Accepted with minor revision
Author Response
We thank the reviewer for the constructive feedback and helpful suggestions, which have contributed to improving the clarity and completeness of our manuscript. Please find below our detailed responses to each comment.
Comment 1: Small sample size and unequal grouping. The study involves only 35 patients, with 14 patients in the regenerative group and 21 in the graft-assisted group. Group imbalance can affect statistical power.
Response: We acknowledge the limitation regarding the small sample size and group imbalance. This study was conducted as a prospective observational analysis within a single specialized center, and patient allocation reflected real-world clinical indications rather than randomization. The unequal group sizes were determined by wound characteristics and surgical eligibility. We have now addressed this limitation in the Discussion section (lines 555 – 560) and emphasized the need for larger, randomized controlled trials to confirm these findings and improve statistical power.
Comment 2: Image legends lack scientific detail: Figure descriptions (figures 7-10) do not specify healing, time intervals, or quantitative assessment. Enhance the figures by adding clinical data in the legends
Response: We appreciate this observation. The legends for Figures 7–10 have been revised to include specific clinical details
Reviewer 2 Report
Comments and Suggestions for Authors
The relevance of the publication is due to the fact that chronic wounds are a constant clinical problem for public health. These injuries require multidisciplinary treatment. Regenerative therapy presents promising strategies to increase wound healing.
The authors indicated that the study was conducted in accordance with the Declaration of Helsinki, approved by the Ethics Committee and informed consent was obtained from all subjects involved in the study.
There are some comments for authors.
- The materials and methods section gives a very brief description of the method for obtaining PRP and autologous fat graft (lines 146-153), but does not indicate how the wound treatment was carried out (lines 115-117).
- Could the authors provide a more detailed description of the methods used to obtain PRP and autologous fat graft or indicate the references?
- For Figures in the Discussion section, the source is not indicated. Figures should be placed after their mention in the text.
- The table name should be placed above the table
Author Response
We thank the reviewer for the constructive feedback and helpful suggestions, which have contributed to improving the clarity and completeness of our manuscript. Please find below our detailed responses to each comment.
Comment 1: The Materials and Methods section gives a very brief description of the method for obtaining PRP and autologous fat graft (lines 146–153), but does not indicate how the wound treatment was carried out (lines 115–117).
Response: Thank you for pointing this out. We have expanded the Materials and Methods section to provide a more detailed description of the wound treatment protocol. This clarification now appears in Section 2.2 of the revised manuscript.
Comment 2: Could the authors provide a more detailed description of the methods used to obtain PRP and autologous fat graft or indicate the references?
Response: We have revised the text Section 2.2 to include a more detailed description of the preparation protocols for PRP and autologous fat grafting.
Platelet-rich plasma (PRP) was prepared using 10 mL Newlife® ACDA tubes, which contain a high concentration of sodium citrate as an anticoagulant. These tubes, equipped with a separation gel, allow efficient partitioning of blood components and preservation of essential bioactive molecules. Blood samples were centrifuged at 3500 revolutions per minute (RPM) for 10 minutes, yielding a high-purity PRP fraction. From each tube, approximately 4–5 mL of PRP was obtained. The final product was administered under sterile conditions by infiltration at an intradermal level at the wound margins and at a depth of 2–3 mm beneath the wound to ensure local delivery directly to the wound microenvironment.
Autologous fat grafting was performed under local anesthesia using Klein solution infiltration at the donor site, typically the inner thigh or abdominal region. Fat was harvested using a 3 mm Mercedes-tip cannula through gentle manual aspiration. The lipoaspirate was then processed using a standardized protocol that included decantation, repeated washing with saline, and mechanical emulsification to reduce particle size. Emulsified fat was passed through 1.2 mm filters to obtain micro-fat, ensuring a more uniform and injectable consistency. The refined graft was subsequently injected into the wound bed and perilesional tissues using a 19-gauge blunt-tip facial fat grafting cannula.
Comment 3: For Figures in the Discussion section, the source is not indicated. Figures should be placed after their mention in the text.
Response:We have corrected the positioning of all figures in the Discussion section to ensure they appear immediately after their first mention in the text. As the images are original and derived from our clinical cases, we have clarified this in the figure captions by adding: “Original clinical image from the authors’ case archive.”
Comment 4: The table name should be placed above the table.
Response: We have revised the formatting of all tables to ensure that the table titles are correctly placed above each table.
Reviewer 3 Report
Comments and Suggestions for Authors
Comments to the authors:
- Weighted network should be provided to estimate the interaction between regenrative medicine and chronic wound management.
- Recent references should be added in the introduction section.
- About grafting, and without the grafting group, the procedure has been missed.
Author Response
We thank the reviewer for the constructive feedback and helpful suggestions, which have contributed to improving the clarity and completeness of our manuscript. Please find below our detailed responses to each comment.
Comment 1: Weighted network should be provided to estimate the interaction between regenrative medicine and chronic wound management.
Response: We appreciate the suggestion. However, as this is a clinical observational study focused on treatment outcomes, a weighted interaction network was not within the scope of our methodology. Nonetheless, we acknowledge the value of such an approach for future research aiming to explore molecular or cellular interactions in regenerative wound healing.
Comment 2: Recent references should be added in the introduction section.
Response: We have reviewed and updated the introduction to include several recent and relevant references published within the last years (ref 13,14,15 and 19), thereby strengthening the background and current relevance of the study.
Comment 3: About grafting, and without the grafting group, the procedure has been missed.
Response: We thank the reviewer for this observation. To address this, we have added a detailed paragraph in the Materials and Methods section describing the split-thickness skin grafting procedure, including graft harvesting technique, fixation, dressing protocol, and postoperative management.
Reviewer 4 Report
Comments and Suggestions for Authors
I would like to recommend this manuscript for publication after minor revision:
- 9 keywords are too many, usually 3-5 are enough;
- The introduction section has too many paragraphs and is fragmented, resulting in weak logic;
- Secondary subheadings such as 2.1, 2.2, 2.3 are too large and not in the original font of the template, making the layout appear out of place;
- Figure 2, Figure 5 and Figure 6 lack vertical axis annotation and units.
- The caption of Figure 2 needs to be described more clearly;
- In Figure 3, the "cm2" on the vertical axis, "2" should be superscript?
Author Response
We thank the reviewer for the constructive feedback and helpful suggestions, which have contributed to improving the clarity and completeness of our manuscript. Please find below our detailed responses to each comment.
Comment 1: 9 keywords are too many, usually 3-5 are enough;
Response: Thank you for your observations. We removed those keywords that were not so relevant to the main idea of the article.
Comment 2: The introduction section has too many paragraphs and is fragmented, resulting in weak logic;
Response: To resolve this issue, we merged several paragraphs so that the text now includes one paragraph for general information, one for the pathophysiology of chronic wounds, one for management, one for PRP, and one for adipose tissue transfer.
Comment 3: Secondary subheadings such as 2.1, 2.2, 2.3 are too large and not in the original font of the template, making the layout appear out of place;
Response: We have corrected the formatting of the secondary subheadings (2.1, 2.2, 2.3) to match the original font and size specified in the journal template.
Comment 4: Figure 2, Figure 5 and Figure 6 lack vertical axis annotation and units.
Response: We have revised Figures 2, 5, and 6 to include appropriate vertical axis annotations and units, ensuring clarity and consistency with journal requirements.
Comment 5: The caption of Figure 2 needs to be described more clearly.
Response: we changed the description to “ Distribution of smoking status among patients treated with skin grafting (graft-assisted group) versus those receiving regenerative therapy alone (regenerative-only group).”
Comment 6: In Figure 3, the "cm2" on the vertical axis, "2" should be superscript?
Response: We have corrected the vertical axis label in Figure 3 by formatting "cm²" with the superscript.
Round 2
Reviewer 2 Report
Comments and Suggestions for Authors
I am grateful to the authors for making changes, but the next questions and comments arose.
- Figure 1. What is the error representation? Is it the standard deviation (SD) [Table 1]? And if Figure 1 duplicates data from Table 1, then it should indicate that BMI is presented as mean in Figure 1.
- Figures 1,3,4. It would have been better if the authors had indicated how they presented the data (mean/median, SD/IQR) and noted whether there was a significant difference.
- [Lines 651-654] It is needed to delete.
- In section 2. Materials and Methods, the authors need to indicate how and where the microbiological analysis was provided.
Author Response
Comment 1: Figure 1. What is the error representation? Is it the standard deviation (SD) [Table 1]? And if Figure 1 duplicates data from Table 1, then it should indicate that BMI is presented as mean in Figure 1.
Response: We thank the reviewer for this valuable observation. In response for Figure 1, the error bars represent the standard deviation (SD), consistent with the data presentation in Table 1. We have clarified this in the figure legend.
Comment 2: Figures 1,3,4. It would have been better if the authors had indicated how they presented the data (mean/median, SD/IQR) and noted whether there was a significant difference.
Response: For Figures 3 and 4, the variables (wound surface area and wound duration) were not normally distributed and are presented as median with interquartile range (IQR). The statistical comparisons were made using the Mann–Whitney U test, and both differences were statistically significant. The figure legends were updated with the information.
Comment 3: [Lines 651-654] It is needed to delete.
Response: Fixed that.
Comment 4: In section 2. Materials and Methods, the authors need to indicate how and where the microbiological analysis was provided.
Response: We have revised the section materials and methods to include detailed information regarding the microbiological analysis.
Reviewer 3 Report
Comments and Suggestions for Authors
The most concerning issues have been addressed.
Author Response
We sincerely thank the reviewer for their thoughtful comments and constructive feedback.